# The Friction of Structurally Modified Isotactic Polypropylene

**DOI:** 10.3390/ma14237462

**Published:** 2021-12-05

**Authors:** Natalia Wierzbicka, Tomasz Sterzyński, Marek Nowicki

**Affiliations:** 1Faculty of Mechanical Engineering, Poznan University of Technology, Piotrowo 3, 60-965 Poznan, Poland; Tomasz.Sterzynski@put.poznan.pl; 2Faculty of Materials Engineering and Technical Physics, Poznan University of Technology, Piotrowo 3, 60-965 Poznan, Poland; marek.nowicki@put.poznan.pl

**Keywords:** isotactic polypropylene, nucleation, friction, surface analysis

## Abstract

The purpose of studies was to analyse an impact of heterogeneous nucleation of modified isotactic polypropylene (iPP) on its tribological properties. The iPP injection molded samples, produced by mold temperature of 20 and 70 °C, were modified with compositions of two nucleating agents (NA’s), DMDBS creating α-form and mixture of pimelic acid with calcium stearate (PACS) forming β–phase of iPP, with a total content 0.2 wt.% of NA’s. A polymorphic character of iPP, with both, monoclinic (α) and pseudo-hexagonal (β) crystalline structures, depending on the NA’s ratio, was verified. The morphology observation, DSC, hardness and tribological measurements as test in reciprocating motion with “pin on flat” method, were realized, followed by microscopic observation (confocal and SEM) of the friction patch track. It was found that Shore hardness rises along with DMBDS content, independent on mold temperature. The friction coefficient (COF) depends on NA’s content and forming temperature—for upper mold temperature (70 °C), its value is higher and more divergently related to NA’s composition, what is not the case by 20 °C mold temperature. The height of friction scratches and the width of patch tracks due to its plastic deformation, as detected by confocal microscopy, are related to heterogeneous nucleation modified structure of iPP.

## 1. Introduction

The polymeric materials belong now-a-days to frequently used in tribological applications. The main advantages are low friction coefficient, easy to produce even very complex sliding elements by injection molding, thus the ability to fabricate industrially applied sliding bearings used nowadays in many branches, such as sport, housekeeping, automotive, medicine etc. The sliding elements are produced mainly in large series; therefore, injection molding presents a very beneficial technique in this case.

The selection of an appropriate polymeric material, shape and dimensions of sliding elements, is a crucial feature by designing polymers for the tribological application. Before selecting the polymeric material for frictional use, the specific application conditions have to be considered.

Numerous polymers and its composites are applied currently by production of sliding elements for different industrial applications. Polymers and modified polymers, polymer blends and/or polymer composites are used now in these applications, and huge literature may be found. Various aspects of sliding are presented recently, such as studies of tribological performance for various polymers, together with role played by the counter face [1], or the results of measurements of coefficient of friction (COF) for different polymers used as slide bearings [2]. The improvement of both, tribological performance and mechanical properties are the topics of papers where the polymer matrix is modified by CNT or graphene added to polyimide [3,4] or polyethylene [5,6,7]. The polymeric bearings are used also in construction operating in extremely demanding conditions [8,9,10]. These are only few examples of recently published studies of polymers applied in tribological designing.

An important circumstance is the establishing of thermal conditions under which the frictional elements will be applied, critical case for majority of polymeric materials. The use of composites with highly resistant polymers filled with thermal conducting powders, such as carbon and/or copper, bronze, aluminum etc. metallic powder allows to overcome the problem of high temperature setting. In this case the coincidence with modification of electrical properties of such composites has to be taken into account.

Mahmoud [11] showed that the best tribological blend was composed of 25% polyethylene and 75% poly(methylmethacrylate) reinforced with bronze, leading to significant decrease in friction and wear. According to Mudradi [12] the use of recycled cast iron powder as a micro-filler for epoxy polymers resulted in reduction of specific wear rate and to decrease in the friction coefficient, compared to pure polymer.

The polymeric sliding bears are especially valuable in cases where relatively low charges are applied, and the continuous lubrication is not required. Some of the polymer- sliders may be water cooled and greased with water cooling lubrication, as it is the case of apparatus and machines working at the food industry and often at agriculture. Such designing’s are also needed by construction of small housekeeping machines, where the use of self-lubricated polymeric materials allows the long utilization without external lubrication.

The relationship between an isotactic polypropylene, used without any additional fillers, however structurally modified by means of specific heterogeneous nucleation, and its tribological behaviors was the aim of our studies. Particularly, the impact of α and β iPP crystalline structure, characterized by various mechanical properties, on the deformation of the external layer of samples charged during the frictional test, was the interest of our experimental research.

Low density, relatively low price, simple processing and high modification ability, by chemical treatment, reaction of composites and by heterogonous nucleation, allowing to create desired application properties, presents main inspirations of industrial use of isotactic polypropylene and its copolymers. Another advantage is a very high popularity of this polymer, presenting pro-ecological advantages such as recycling ability with all commonly used processing techniques. These effects are described in literature, such as mechanical properties and reusability of printed circuit boards, [13] formation of materials from recycled PP blends with TPE [14] or recycled blends of polyolefins [15], use of recycled polypropylene for composites with natural fibers [16], and the reuse of industrial PP products [17]. It has to be stressed that polypropylene may easily be recycled at the end of lifetime, thus offers a limited devastation of the environment.

The early information’s concerning structural modification of iPP by nucleation induced crystallization were published by Beck [18] and Binsbergen [19]; suggesting a theoretical explanation of these effect [20] based on the surface energy. Wittman and Lotz [21,22] have proposed the theory of epitaxial growth to explain the specific interaction between the nucleating agent and the crystalizing polymer.

The heterogeneous nucleation modification of the crystalline structure is a very effective and well-known method, allowing production of iPP products with desired properties. Lovinger et al. [23] presented a method of preparation of β pseudo hexagonal crystalline forms in polypropylene in temperature gradient. A huge bibliography may be found concerning the use of a small quantity of various low molecular additives as nucleating agents resulting in specific iPP crystallization (NA’s) [24,25,26,27,28,29]. Two main effects may be awaited if the iPP is modified by using the NA’s, the first one is characterized by an increase of the crystallization temperature Tg, allowing to shorten the processing cycles, the second one may lead to specific creation of various crystalline forms. Recently a review of known organic and nonorganic additives, playing a role of nucleation agents, was presented by Gahleitner et.al. [30].

The use of specific NA’s, leading to an increase of nucleation density, thus to a morphology with small homogeneously distributed spherulites, in the crystalline polymer was published in [31,32,33]. Instead of creation of homo crystallization centers by local overcooling, a high number of regularly distributed NA’s particles leads to formation of heterogeneous nucleation centers, followed by its growth, which happens in a temperature higher comparing with the neat polymer. As the molten polymer begins to crystallize/solidify in higher temperature a significantly shorten cycles time is observed, an effect which is largely notable and often applied by industrial processing.

In addition to the increase of crystallization temperature, the heterogeneous nucleation leads to transformation of the crystalline structure and consequently of the properties measured even on the nano level [34,35,36,37,38]. This effect concerns not only the iPP homopolymer, but also its copolymers with PE [25,30,38].

A crystalline phase transition may be observed if appropriate additives are applied. The isotactic polypropylene may crystallize in three crystalline phases, i.e., in monoclinic α–phase, hexagonal β–phase and rarely in triclinic γ–phase [39,40,41], where the physical properties, such as mechanical strength, choc resistance, stability and even transparency are strongly the crystalline form dependent. Concerning the specific properties for the technical/industrial uses, basically only the monoclinic and hexagonal crystalline forms of iPP have found an application nowadays, because the iPP crystalized in the monoclinic form is characterized by high modulus, thus lower deformability and brittle type of break comparing with pseudo-hexagonal structure of β iPP. Another important property is an improved transparency of α–phase iPP, comparing with β–form samples, effects which schematically may be seen in Table 1.

As mentioned before the use of low molecular additives with nucleation capacity leads to morphology with very small spherulites, even in a cylindrical form [42]. As the type and dimension of the spherulitical morphology significantly influence the polymer properties, the same nucleation density was applied in all samples, i.e., an equal quantity of NA’s additives was merged into the polymeric matrix.

Our interest was to investigate polymeric samples with comparable morphology, but with various well defined crystal structures, therefore the same total quantity of 0.2 wt.% of combination of specific α and β NA’s was used in all samples. Taking into account different iPP crystallization ability of both NA’s, the creation of monoclinic (α) and pseudo-hexagonal (β) crystalline structures was predictable.

## 2. Materials and Methods

### 2.1. Plan of Experiments

The basic mechanical properties of an isotactic polypropylene (iPP) modified by specific heterogeneous nucleation, were evaluated as a part of this study. The surface texture and an impact of structural modification on tribological behaviors were completed as a second part of the experiments. These tests include the tribological investigations, such as determination of friction coefficient and analysis of pathway deformation.

### 2.2. Materials

An isotactic polypropylene Moplen HP500 N (Basell Orlen, Plock, Poland), a non-nucleated injection molding grade, without nucleating agents was used as a matrix polymer. The characteristic values of this polymer are following: MFR (210 °C, 2.16 kg) = 12 g/10 min; MFV (210 °C, 2.16 kg) = 16 cm^3^/10 min, density 0.90 g/cm^3^, HDT = 97 °C and the flexural modulus Eb = 1.48 GPa.

The 1,3:2,4- Bis (3,4-dimetylobenzylideno) sorbitol (DMDBS) Millad 3988 produced by Milliken Chemical (Gent, Belgium) was used as specific α nucleation agent, leading to creation of monoclinic crystallographic form of iPP.

According to our previous studies and to a number of publications [43,44,45,46,47,48,49], a 1:1 mixture of pimelic acid and calcium stearate (PACS) (both delivered by Sigma-Aldrich (Darmstadt, Germany)) in a powder form, was used as specific nucleating agent, leading to formation of the β hexagonal phase of iPP.

The composition of samples is presented in Table 2.

### 2.3. Samples Preparation

The master batches (MB) of α and β nucleating agents, with a concentration of 0.5 wt.% each, were homogenized using a twin-screw extruder ZAMAK (Zamak Mercator, Skawina, Poland), operating at temperature 150 °C to 200 °C along the barrel, followed by solution of the MB with iPP to a concentration of 0.2 wt.%.

Regarding our primary experiments [50,51], the most suitable content is 0.2 wt.% of NA’s in the iPP matrix, leading to an improvement of crystallization process and properties. On this way both, a beneficial increase of the crystallization temperature and creation of the demanded crystal phase may be achieved. Such a high content of NA’s creates a stable small-spherulitical morphology of iPP, which may clearly be observed by optical microscopy [52,53]. Therefore, in our investigations the total content of nucleating agents in the polymeric matrix was fixed for all samples, equal to 0.2 wt.%.

The iPP matrix modified by α and β specific nucleating agents (DMDBS and PACS, respectively) with relative concentration of both NA’s 1:0, 1:3, 1:1, 3:1 and 0:1 was examined together with neat iPP samples. Two series of nucleation modified iPP samples were investigated, produced by two injection molding with mold temperature of 20 °C and 70 °C. Due to literature and our earlier experiences, it was ascertained that even if a proper β –NA has been applied, by fast cooling at the mold, only α–phase crystals are usually achieved. This structural effect was indicated and explained by Varga [52] who found that the Tα,β temperature of 144 °C is critical for selective creation of both crystal phases in an isotactic polypropylene.

The samples were produced using the Battenfeld PLUS 350 (Battenfeld, Bad Oeynhausen, Germany) injection molding machine, operating by barrel temperature 210–220–220 °C, injection pressure 26 MPa, packing pressure 21.5 MPa, packing time 20/23 s and cooling time 50 s, at the dimension of 100 × 90 × 2 mm^3^.

### 2.4. Polarized Microscopic Observation

To detect the spherulitical morphology the polarized microscopic observations (POM) were realized in transmission mode. Samples in a form of slights with a thickness between 10 and 20 × 10^−6^ m were cut by using a Leica Microtome (Leica Microsystem, Buffalo Grove, IL, USA) on the cross section of the PP plates.

### 2.5. Differential Scanning Calorimetry

The differential scanning calorimetry (DSC) measurements were realized to identify the existence of both, the monoclinic and the hexagonal structure, together with the specific melting and crystallization temperature. The measurements were performed using the Netzsch DSC 204 F1 Phoenix (Netzsch, Selb, Germany) with an intercooler, operating at double heating/cooling mode, at the temperature range between 20 °C and 240 °C. The heating and cooling rate was equal to 10 °C per minutes and the average mass of samples was about 5 mg.

### 2.6. Wide Angle X-Ray Scattering

The X-ray diffraction analysis, with the aim to determine the type of the crystal structure of nucleation modified PP samples, was carried out by means of X-ray diffractometer Seifert FPM URD-6 (Rich Seifert-FPM, Berlin, Germany), operating by 40 kV and 20 mA, with a Bragg-Brentan geometry, at scanning angle between 10 and 40°, using the Cu_kα_ radiation in reflection mode. The X-ray wavelength was 0.14518 nm.

The k-value giving a quantitative information about the hexagonal *β* phase content was evaluated by means of Turner—Jones et al. equation in a form [54] (1):(1)$$k=\frac{I_{\beta 1}}{I_{\beta 1}+I_{\alpha 1}+I_{\alpha 2}+I_{\alpha 3}}\cdot 100\%$$
where,

*I*—diffraction intensity by following crystallographic planes:

$I_{\beta 1}$—(300) β phase,

$I_{\alpha 1}$—(110) α phase,

$I_{\alpha 2}$—(040) α phase,

$I_{\alpha 3}$—(130) α phase.

I(300) is the diffraction intensity by Bragg diffraction angle 2θ = 16° characteristic for the hexagonal *β* phase of polypropylene, corresponding to crystallographic plane with Muller indices (300).

### 2.7. Hardness Test

The hardness of the samples was measured by means of Shore Durometer (Sauter GmbH, Wutöschingen, Germany) where the measurements were performed by a penetration distance of about 5 mm one to each other on the samples surface. The average value of 10 measurements was presented as the result of measurements.

### 2.8. Tribological Tests

Tribological tests were realized by using the tribological tester BRUKER UMT2 Tribolab (Bruker Corporation, Billerica, MA, USA), operating at the pin-on-flat configuration under reciprocation displacement.

Samples were examined in accordance with the applicable standards (ASTM G99). The pin, a ball made of 1.2378 steel (a diameter of 6.35 mm and a roughness of Ra = 0.6 μm) was held in continuous sliding contact with the composite specimen, according to methodology described by Czapczyk et al. [55].

The tribological test with a duration of 60 min and an amplitude 40 mm by maximal velocity 25 mm/s was realized. The tests were carried out for three values of force, respectively, 5 N, 25 N and 45 N. For further experiments, the middle force value was chosen due to the similar values of the coefficient of friction for all three measurements. Additionally, the measurements with the force of 25 N were most stable. Coefficient of friction was evaluated by expression (2) [56,57]:*μ* = (*A_r τ*)/*W*(2)
where *μ* is the friction coefficient, *A_r* is real contact area, *τ* is effective shear strength of contacts, and *W* is the normal load (*N*).

For testing with this method, the tribotester was equipped with a fixed table, in which the sample is clamped with a vice. It allows an immobilization of the sample and the prevent it from moving in any way, during the measurement.

The measurements of each composite were carried out at least three parallel experiments, to ensure a relevant statistical evaluation. The average value of friction coefficient is presented with corresponding standard deviations. The tests were conducted in dry sliding conditions and at room temperature. Before investigation the samples were cleaned in the deionized water bath for 20 min.

### 2.9. Surface Morphology and Topography Tests

The deformation of the dynamically charged sliding surface after tribological test was analyzed by means of scanning electron microscopy (SEM) observations, using the FEI Quanta 250 FEG microscope (FEI Company, Hillsboro, OR, USA). The accelerating voltage was 10 kV, images were recorded in the secondary electron mode. The samples were not covered and the measurement was performed in a low vacuum mode (about 70 Pa).

### 2.10. Confocal Microscopy

Topography measurements were carried out with the Olympus LEXT 4100 laser scanning microscope (Olympus, Hamburg, Germany). The surface of the sample was scanned with laser light (405 nm), which allowed to collect the topography image and measure the roughness of similar-sized areas analyzed always using the same lens/magnification. The Root Mean Square Roughness are presented as a roughness parameter Root mean square height (*Sq*). This parameter expands the profile (line roughness) parameter Rq three dimensionally. It represents the root mean square for *Z*(*x*, *y*) within the evaluation area (3) [58]:(3)$$Sq=\sqrt{\frac{1}{A}\overset{n}{\underset{A}{\iint }}Z^{2}\left(x,y\right)\;dxdy}$$

## 3. Results and Discussion

### 3.1. Spherulitical Morphology

The microscopic observations in polarized light microscopy (POM) were made using Nikon Eclipse E400 microscope equipped with an Opta-Tech digital camera (OPTA-TECH, Warsaw, Poland) for three types of samples, namely neat iPP, iPP with 0.2 wt.% of PACS and iPP with 0.2 wt.% of DMDBS produced by both mold temperature, 20 °C and 70 °C, are presented on Figure 1.

As it follows from the Figure 1, a significant influence of heterogeneous nucleation on iPP morphology is clearly visible. The neat iPP samples are characterized by well-developed spherulites, where only a low influence of the mold temperature, in a form of slightly better developed morphology by higher mold temperature, may be observed. The spherulitical morphology of iPP nucleated with 0.2 wt.% of both NA’s, i.e., PACS and DMDBS, is characterized by homogeneous distribution of very small spherulites, even in cylindrical form, an effect in agreement with many publications [41,42,43,44,45,46,47,48,49,50,51,52,53]. Once more, by higher mold temperature slightly larger spherulites are developed by cooling induced crystallization.

### 3.2. Melting and Crystallization Temperature by DSC

The melting and crystallization temperature of non-modified and modified iPP samples, measured by differential scanning calorimetry, are presented in Table 3.

These results were recorded during second DSC run, as by first heating and cooling not only the structure of investigated polymer, but also its thermal and mechanical history are reflected, such as a specific ordering of the macromolecular network in a form of macromolecular orientation. Thus, for the structure determination more credible is the second run, realized for samples crystalized in protective atmosphere in controlled constant cooling rate. The DSC results, i.e., the melting and crystallization temperature, characteristic for both iPP crystal phases, measured during second DSC run, are presented in Table 3. As it follows from the Table 3, for samples modified with β-phase NA a characteristic double melting temperature was observed. According to [52] the first peak at the temperature about 151 °C is attributed to melting of the hexagonal—β iPP crystalline phase, followed by iPP recrystallization in monoclinic form, and subsequently melting of the monoclinic—α-phase of iPP, observed at the temperature at the range of 167 °C. The double melting peaks was not observed in the case of DMDBS and 3 D1 P samples, signifying that in theses samples only α-form crystals exist.

By DSC cooling run, a shift of crystallization temperature, depending on the composition of the NA’s may be seen. A significantly higher crystallization temperature was observed relatively to the content of the α–monoclinic NA, due to superior nucleation efficiency of this additive [50,51].

### 3.3. The Polymorphic Crystalline Structure by WAXS

The existence and coexistence of both, monoclinic and hexagonal crystallographic phases was confirmed by Wide-Angle-X ray Scattering (WAXS) measurements, signifying the presence of a polymorphic effect [35,44,59,60] as shown by Garbarczyk and others. The value of k, characteristic for pseudo-hexagonal β-phase content, evaluated by using Turner-Jones equation [54], for samples modified with DMDBS and PACS, are presented on Figure 2. The value of k = 0 signifies the existence only of an α monoclinic crystal form, and the k = 1 denotes the fully developed β-phase in polypropylene. The k –value (0 < k < 1) is a proof of the polymorphism, characteristic for a coexistence of both crystallographic forms. For the neat iPP samples crystallized by mold temperature of 20 °C, as well as for samples containing only the DMDBS and the majority (3:1 DMDBS:PACS) of this NA, only the monoclinic α-phase was detected, an observation concerning both mold temperature, i.e., 20 °C and 70 °C. For other samples an increase of the k-value (β-phase contain) is related with growing concentration of PACS nucleating agent. A certain k-value observed in the case of neat iPP crystallized by injection mold temperature of 70 °C may be explained by effect proposed by Varga [52,53]; a possibility of β- phase creation by high shearing in molten state, what is the case of injection molding. This effect may be observed by relatively high mold temperature, where a lower cooling rate, let to create this crystalline structure at the temperature above the Tα/β transition state.

### 3.4. Hardness Measurements

The hardness values of investigated samples are presented in Figure 3. An increase of the Shore values, depending on the DMDBS contain is clearly noticeable, particularly in the case of higher mold temperature. This tendency was observed for all samples, with an exclusion of iPP modified entirely with PACS, where the hardness values are lower and seems to be less mold temperature dependent.

Another observation was a slight decrease of hardness, for very low DMDBS content (Figure 4) with simultaneous increase of the relative content of β–form NA, (comp. Table 1) responsible for the creation of iPP with lower hardness and modulus as well as improved deformability. On the contrary for higher DMDBS concentration an increase of the Shore hardness values was noted. These effects are in agreement with general assumptions that for the α–phase polypropylene, superior modulus and hardness values are usually observed, as it was also shown by Aboulfaraj et al. [61] in relation to specific design of lamellas of α and β spherulites.

### 3.5. Evaluation of Tribological Properties

The charts of changes in the friction coefficient were obtained automatically from tribological tests using the computer software attached to the BRUKER UMT TriboLab device. For all samples the value of the friction coefficient was stabilized after certain patch way (Figure 5).

A tendency of sliding distance dependent stabilization of the steady state COF values for PP samples with various NA’s composition, produced by mold temperature 20 °C and 70 °C, may be seen on Figure 5.

As it follows from Figure 5, at the beginning of the measurement the coefficient of friction is relatively low and the grows to finally achieve a steady state. This behavior is typical for polymers [62,63]. Due to the adhesion of the polymer to the steel counter-sample, a thin layer of polymer (so-called transfer film) can be deposited on the opposite sample [64]. All of those factors contribute to the changes of the coefficient of friction in function of sliding distance.

Analyzing the changes of Coefficient of friction for all samples (Figure 5), it can be seen that for the samples produced at lower mold temperature the stabilization occurs after a short time. In the case of higher mold temperature, the values of COF are noticeable higher and the stabilization time is generally longer. The differences in stabilization time may result from changes of the outer layer properties, such as more intense friction induced deformation of the surface, as it will be shown later by discussion of confocal observation of the patch tracks.

Additionally, when analyzing the course of individual measurements (Figure 5), it can be noticed that in the case of iPP samples without NA’s, the friction coefficient quickly reached the maximum value, while in the case of other materials this process was extended.

The selected values of the friction coefficient, as examples of measurements by different forces are shown in Figure 6. For samples produced by mold temperature of 20 °C the COF reveals generally lower values, comparing with higher mold temperature. The COF for these series of samples (formed by 20 °C mold temperature) reveal values at the range between 0.24 and 0.26, and less influenced by the structure modification. This observation may be explained by an effect proposed by Varga and confirmed by our measurements, i.e., a tendency of α– form crystal structure creation, less dependent on the NA’s composition. In the case of higher mold temperature (70 °C) the COF values are in the range between 0.25 and 0.33, i.e., higher and also more differentiated, thus dependent on the crystal structure formed by using of selective α and β NA’s.

Analyzing the graph (Figure 7), presenting the value of friction coefficient as a function of the DMBDS, a constant value of COF was observed for a low content of this NA up to 0.1 wt.%. A certain decrease of the COF value of about 40% was noted for the DMDBS content of 0.15 and 0.2 wt.%. These changes are similar for both temperatures.

### 3.6. Scanning Electron Microscopy and Confocal Microscopy Observation of Friction Area

The friction area was observed by SEM and by confocal microscopy. Furthermore, the roughness after friction test for the sliding patch is presented based on the confocal observations. On the SEM photos a deformational character of the surface of the samples, due to the plastic-like displacement of the skin layer, may be observed. This effect is related to both, the mold temperature by samples production and physical modification of the polymer. Comparing this effect for three types of samples (PP neat polymer, iPP modified with PACS and iPP modified with DMDBS) always a more intense deformation of the samples surface for higher mold temperature was noted. For twin- nucleated samples (formed by 20 °C mold temperature) this effect is also visible, where the intensity of plastic surface deformation is DMDBS content dependent.

The SEM observations are confirmed by results of confocal microscopy, where in addition to surface observation also data’s concerning the vertical displacement of the material were registered. These friction-induced vertical displacements of iPP are shown in a form of 3 D graphs on Table 4, and in numerical values of roughness Sq of the surface before the tests (1) and as scratch formed during the friction test, in Table 5. The width of the friction way observed by confocal microscopy, is presented graphically in comparison with hardness measurements on Figure 8.

For all samples, except neat iPP 20, an increase in the roughness value in the area of the scratch, formed after the friction test, compared to the original surface was observed. Two distinct groups of samples were observed: the first one (PP70, PACS20, PACS70, DMBDS20) with a roughness value between 100 and 250%, and the second (DMBDS70, 1D1P20, 1D3P20, 3D1P20) with corresponding roughness between 600 and 800%. These outcomes are in agreement with a former analysis of the SEM observations, where also for the same group of samples more intense displacement of the friction surface was observed. The possible explanation may originate by a higher deformation susceptibility of the surface where the α—modification leads to somehow higher hardness, and consequently to lower relaxation of local pin induced deformation of the surface during the frictional test.

On Figure 8 comparison between hardness of the samples and the width of the patch track, measured by the confocal microscopy, may be seen. As discussed already before, a correlation between the specific nucleation induced structure and the hardness value is observed. The width of the tracks produced by friction test are at the range between 500 and 1200 mm, where the highest values correspond to the hexagonal structure of iPP, i.e., to the polymer with the biggest potential plastic deformability. It may be assumed that these samples, under friction load, undergo a plastic deformation resulting in a superior width and lowest depth of the friction tracks. Consequently, the coefficient of fiction of these samples is slightly higher, particularly for the material produced by injection molding with a mold temperature of 70 °C, leading to higher content of the β–phase.

## 4. Conclusions

The presence of both, monoclinic and hexagonal crystallographic phases was confirmed, signifying the existence of a polymorphic effect. Hardness increases along with DMDBS content where the changes are similar for both mold temperatures. The value of the coefficient of friction depends on the nucleation agent content and the mold temperature. For the 20 °C, lower deformation of the patch track surface occurs and the content of individual additives do not significantly affect its value. However, in the case of 70 °C, these values are higher and more divergent depending on the composition.

The use of 0.15 wt.% of DMBDS, by both mold temperatures, 20 °C and 70 °C, leads to higher resistance of the misshaping of the patch track. This effect is related to both, the mold temperature by samples production and to physical modification of the polymer. Comparing this effect for three types of samples (PP, PACS—modified and DMDBS—modified) always a more intense deformation of the samples surface for higher mold temperature was noted. For twin-nucleated samples (formed by 20 °C mold temperature) this effect is also visible, where the intensity of plastic deformation is slightly DMDBS content dependent.

Based on the mechanical tests and optical observations of the structurally modified iPP frictionally tested, samples following conclusions may be formulated.

By selective modification of iPP with gradually varying content of α and β nucleating agents samples with various content of hexagonal β phase may be produced.The specific structure creation in an isotactic polypropylene is followed by changes of the mechanical properties, where usually higher hardness for α—modified samples may be observed.By higher mold temperature, resulting in a lower cooling rate, thus by more nucleation induced structures of iPP, somehow higher values of COF were noted.Regarding the geometrical profile of the patch tracks, as observed by confocal microscopy, the height of friction induced scratches is related to the heterogeneous nucleation modified structure of iPP.The width of the tracks, due to the friction provoked plastic deformation of the modified iPP, appears to be dependent on the heterogeneous nucleation induced structure.

## Figures and Tables

**Figure 1 materials-14-07462-f001:**
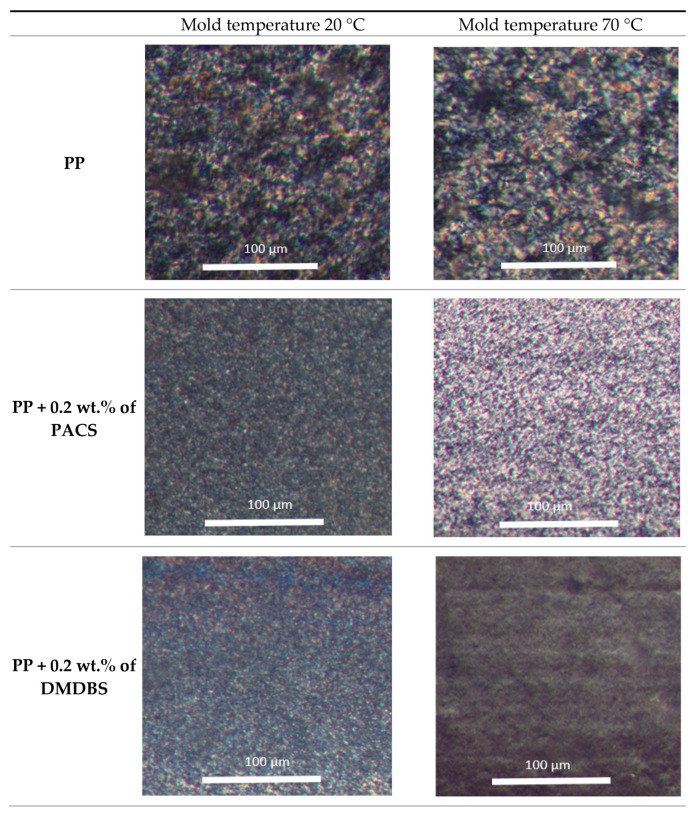
The microscopic observations by polarized light (POM).

**Figure 2 materials-14-07462-f002:**
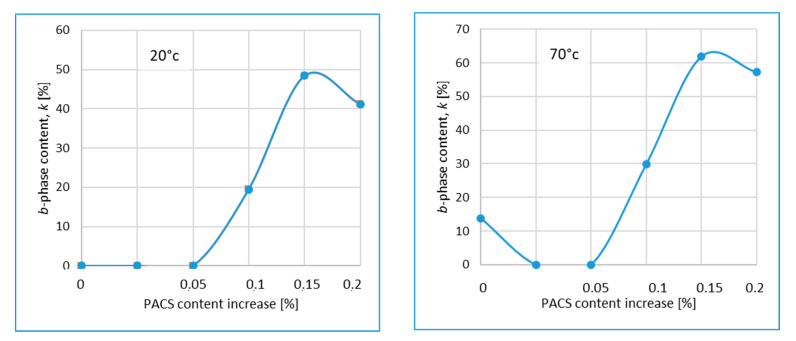
The value of k for samples modified with DMDBS and PACS for 20 °C and 70 °C.

**Figure 3 materials-14-07462-f003:**
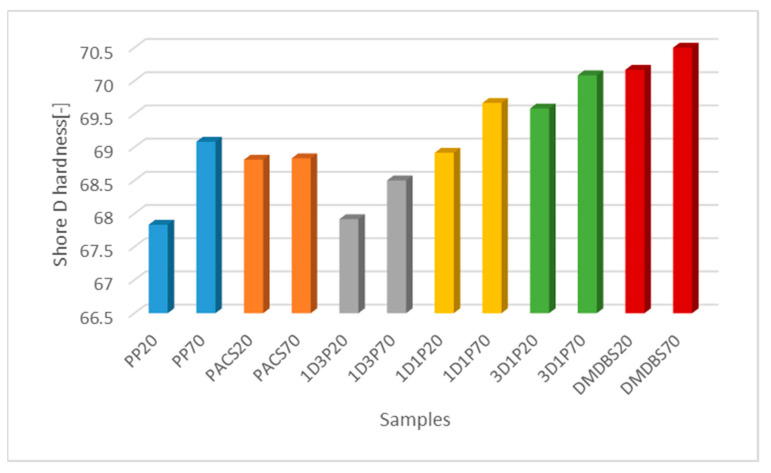
The Shore D hardness of the PP samples with various NA’s composition produced by mold temperature of 20 °C and 70 °C.

**Figure 4 materials-14-07462-f004:**
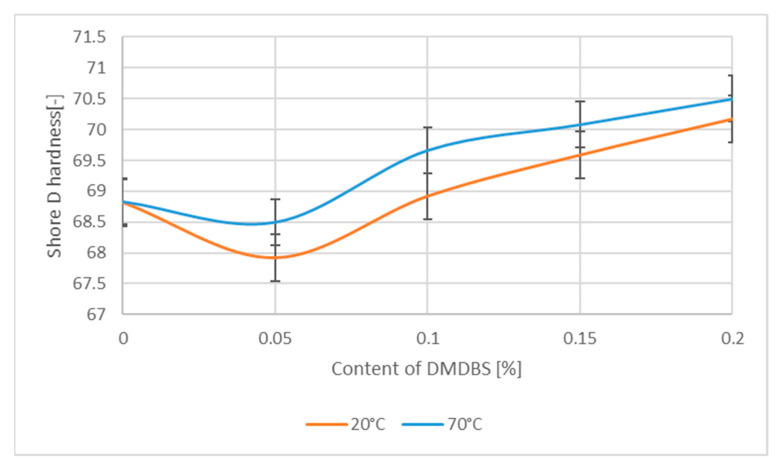
The Shore D hardness as a function of the DMDBS contain in the PP samples.

**Figure 5 materials-14-07462-f005:**
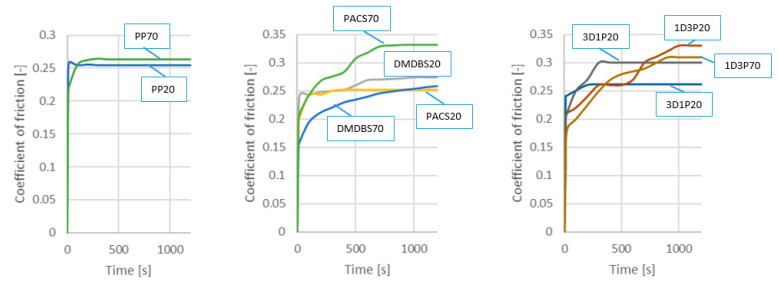
The coefficient of friction as a function of the sliding distance for the samples 20 °C and 70 °C for the normal load of 25 N.

**Figure 6 materials-14-07462-f006:**
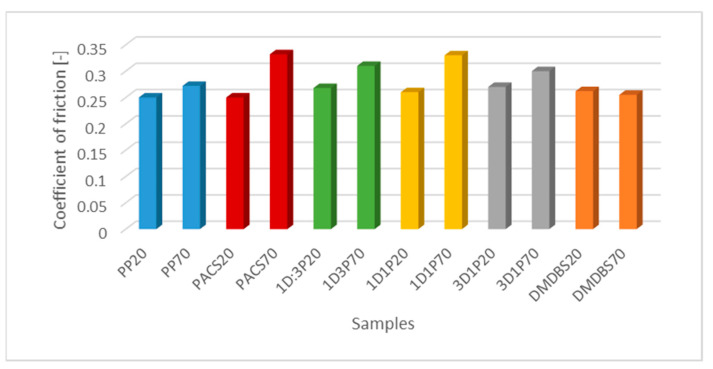
The steady state values of the coefficient of friction of the PP samples with various NA’s composition produced by two mold temperature 20 °C and 70 °C.

**Figure 7 materials-14-07462-f007:**
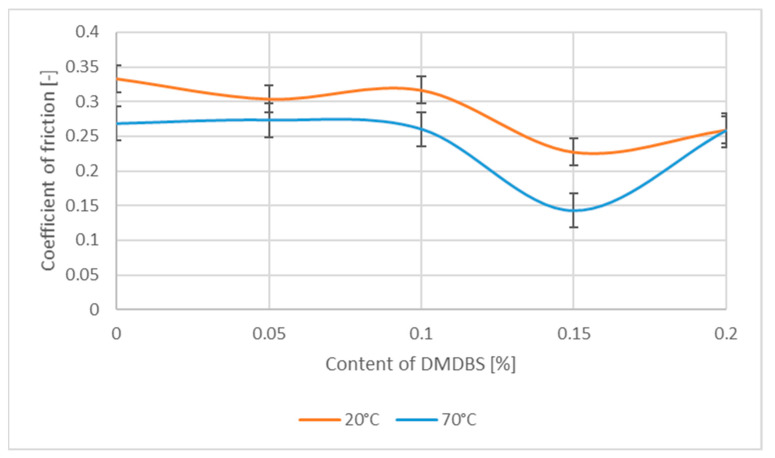
The coefficient of friction as a function of the DMDBS contain in the PP samples.

**Figure 8 materials-14-07462-f008:**
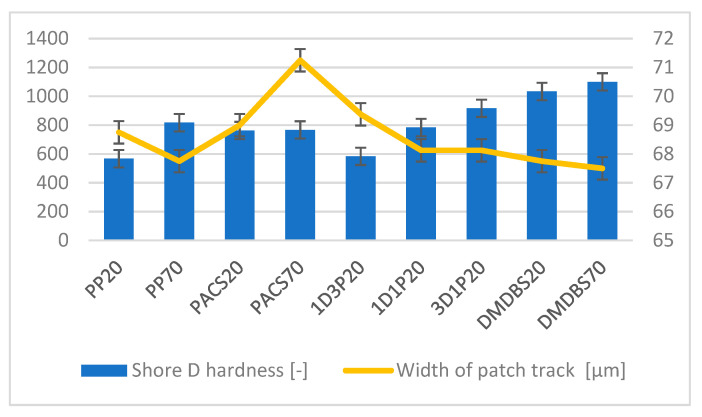
Comparison between hardness of the samples and the width of the patch track.

**Table 1 materials-14-07462-t001:** Mechanical properties and transparency of α–phase iPP comparing with β–form samples.

Specific Property	Monoclinic iPP α	Pseudohexagonal iPP β
Tensile modulus of elasticity	HIGH	LOW
Tensile elongation at break	LOW	HIGH
Impact resistance	LOW	HIGH
Optical transparency	HIGH	LOW

**Table 2 materials-14-07462-t002:** Composition of iPP nucleated samples. 20 and 70 °C correspond to injection mold temperature.

Notation	Total Concentration of Additives [wt.%]	Content of NA [wt.%]	
		DMDBS	PACS
**PP 20**	**0**	-	-
**DMDBS20**	**0.2**	0.2	-
**3 D1 P20**	**0.2**	0.15	0.05
**1 D1 P20**	**0.2**	0.1	0.1
**1 D3 P20**	**0.2**	0.05	0.15
**PACS20**	**0.2**	-	0.2
**PP70**	**0**	-	-
**DMDBS70**	**0.2**	0.2	-
**3 D1 P70**	**0.2**	0.15	0.05
**1 D1 P70**	**0.2**	0.1	0.1
**1 D3 P70**	**0.2**	0.05	0.15
**PACS70**	**0.2**	-	0.2

**Table 3 materials-14-07462-t003:** The melting and crystallization temperature characteristic for both iPP crystal phases.

Samples	Crystallization Temperature $T_{kr}\;[{}^{\circ}C]$	Melting $T_{t}\;[{}^{\circ}C],\;\mathrm{First}/\mathrm{Second}\;\mathrm{pic}\;(\beta /\alpha )$
PP20	116	148.1/164.1
DMDBS 20	128.4	-/164.8
3 D1 P20	128.4	-/164.6
1 D1 P 20	123.3	150.7/164.5
1 D3 P20	123.7	150.8/167.5
PACS20	123.9	151.0/168.4
PP70	116.3	148.0/163.7
DMDBS70	128.2	-/165.0
3 D1 P70	128.4	-/164.4
1 D1 P70	123.5	150.6/163.1
1 D3 P70	123.5	151.1/167.6
PACS70	123.8	151.0/167.6

**Table 4 materials-14-07462-t004:** Comparison of friction area observed by Scanning Electron Microscopy and confocal microscopy.

Samples	Scanning Electron Microscopy	Confocal Microscopy	
PP20	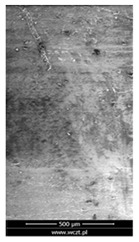	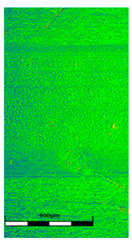	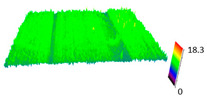
PP70	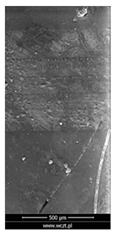	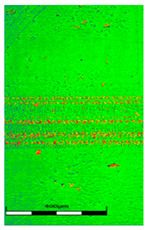	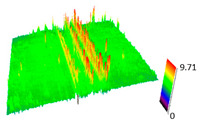
PP + PACS20	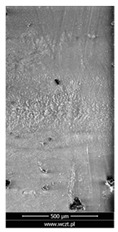	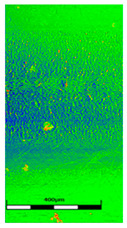	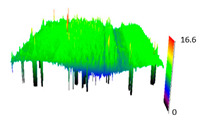
PP + PACS70	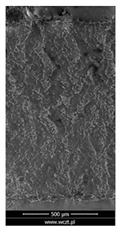	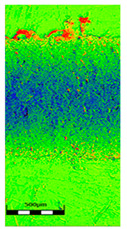	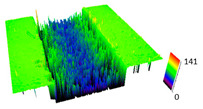
PP + DMDBS20	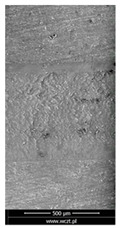	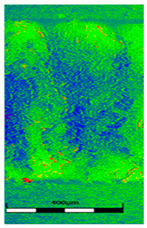	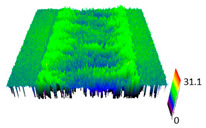
PP + DMDBS70	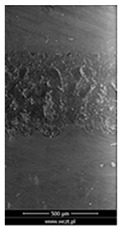	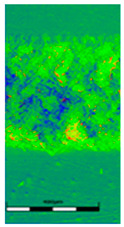	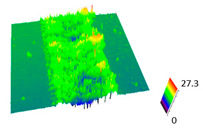
PP 1 D1 P20	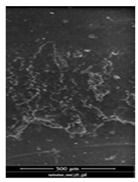	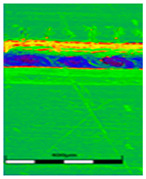	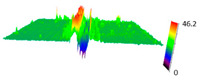
PP 1 D3 P20	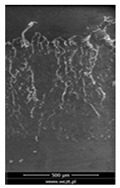	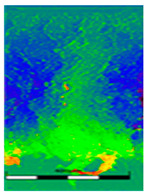	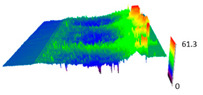
PP 3 D1 P20	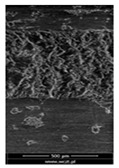	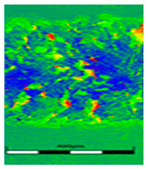	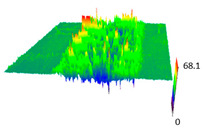

**Table 5 materials-14-07462-t005:** The roughness parameter of the surface before the tests (Sq1) and as scratch formed during the friction test (Sq2).

Sample	Sq1 [µm]	Sq2 [µm]	ΔSq
PP20	2.03	1.83	−0.2
PP70	0.54	1.75	1.21
PACS20	0.73	2.18	1.45
PACS70	1.81	5.78	3.97
DMBDS20	1.71	3.64	1.92
DMBDS70	0.53	4.48	3.95
1 D1 P20	0.80	7.19	6.38
1 D3 P20	0.72	5.28	4.56
3 D1 P20	0.80	7.19	6.38

## Data Availability

Data sharing not applicable.

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
