# Peer review of "The Friction of Structurally Modified Isotactic Polypropylene"

_materials, 2021, doi:10.3390/ma14237462_

Round 1

Reviewer 1 Report

This study presents the tribological properties of heterogeneous nucleated iPP processed by injection molding. Results seem to be good, but it is difficult to find the originality in such a studied material as iPP. English should be improved, all the manuscript is written in a very complex manner, wording needs to be simplified. Also grammar should be checked and corrected. I think that authors should improve the quality of the paper that is not always clear. Technical quality is good, but not outstanding, and my concern is about novelty.

Please find below some suggestions and remarks:

  • Introduction section is overly extensive and the originality of the work is not clearly stated. I think that before publication, authors should improve this section by reducing it and emphasizing the novelty of the work.
  • Section 2.1 Plan of the experiment is unnecessary. This can be explained – as one or two sentences – in the introduction, when the aim of the work is stated. Figure 1 is a nice graphical abstract, but it is not necessary in the manuscript.
  • Section 3. Table 3 should be included as a figure, not a table
  • Section 3, page 10. DSC second run is mentioned in line 316, but explained in line 20. Sentence of lines 315 and 316 should be placed after: “… of protective atmosphere.”(line 325).
  • Section 3 is very difficult to understand. This section should be organized: first findings, then explanations.  

Author Response

Dear Reviewer,

Thank you for your remarks and comments which will help us to improve the scientific level of our paper.

We have introduced corrections according to your comments. Below please find the replies to comments:

  1. This study presents the tribological properties of heterogeneous nucleated iPP processed by injection molding. Results seem to be good, but it is difficult to find the originality in such a studied material as iPP. English should be improved, all the manuscript is written in a very complex manner, wording needs to be simplified. Also grammar should be checked and corrected. I think that authors should improve the quality of the paper that is not always clear. Technical quality is good, but not outstanding, and my concern is about novelty.

ANSWER: Although an isotactic polypropylene is used for many years, but it is still a very important and widely used polymer. In 1980s the replacement of polyolefins by so called hi-tec and commodity polymers was planned, with  a prevision of significant lowering of PP and PE production. On the contrary, at the end of the twenties century, due to economic crises and ecological movements, this idea was fully stopped, leading consequently to a significant increase of PP and PE fabrication. Beside relatively low prices of these polymers, varied modification possibilities led to  an essential growth of the PP production, which present today up to 20 % of the total world polymer production (about 70 billion ton every year) (see reports from Bayer AG).

This is the reason today, why a  number of various modification induced improvements  of PP properties, for desired application, may be observed and followed in hundreds of publications. According to Scopus more than 300 papers were published (in 2021 about 20 papers) reporting only the PP - friction experiments. In all 2021 papers, PP was modified by creating composites with various additives, thus producing materials difficult to be recycled. Our idea was to create new desired properties by structural modification, without any typical additives, thus PP construction material for friction application, which may be fully recycled after the end of the life  time. 

  1. Introduction section is overly extensive and the originality of the work is not clearly stated. I think that before publication, authors should improve this section by reducing it and emphasizing the novelty of the work.

ANSWER: We have reduced the introduction part as far it was imaginable to maintain its informative character from both sides, i.e. characterization of structurally modified polymer and specificity of the friction research. The novelty of our research is the analysis of friction and friction induced deformation of a run way of a semi-crystalline polymer, with purposefully modified structure by heterogeneous nucleation, without any fillers, thus using a material which may be fully recycled.  

  1. Section 2.1 Plan of the experiment is unnecessary. This can be explained – as one or two sentences – in the introduction, when the aim of the work is stated. Figure 1 is a nice graphical abstract, but it is not necessary in the manuscript. - CORRECTED
  1. Section 3. Table 3 should be included as a figure, not a table - CORRECTED
  1. Section 3, page 10. DSC second run is mentioned in line 316, but explained in line 20. Sentence of lines 315 and 316 should be placed after: “… of protective atmosphere.”(line 325). – CORRECTED
  1. Section 3 is very difficult to understand. This section should be organized: first findings, then explanations.

ANSWER: We have fully reorganized the section 3, expecting to make it easier to follow and to understand.

I hope that these changes are sufficient and will positively influence the evaluation of the manuscript.

I am enclosing the Manuscript.

Yours faithfully,
Natalia Wierzbicka

Reviewer 2 Report

4-5 lines long, mostly incomprehensible or incorrect sentences should be rewritten everywhere, or unnecessary connecting texts should be deleted (e.g. lines 64-70)

- Sentence in line 103-104 ??

Table 1.: Unlucky to use the sliding/moving arrows, other markings are required

-Line 169: ... very low content of additives ..... this is not a definition

- “contain” word is a verb and not a noun (line 189, 194)

- figure 2: bad figure, contrary to the explanation of line 250. The pin is actually a steel sphere.

figure 3. Not relevant to the content of the article, can be deleted

-Table 3 photos: In the figures, the parts explained in the text later should be marked with some emphasis, identifiable by the textual explanation.

- Figures 5 and 6. A more detailed material structure explanation is required for the observed hardness change.

- Friction curves: analysis of running-in and steady state friction is required separately. The phenomenon of running-in is well known in polymer tribology: the polymer transfer film characterizes the formation on the metal surface and the attainment of its equilibrium state. It is strongly related to the surface energy and adhesion of the polymer.

- Friction analysis is not possible at least, the simplest model e.g. Without regard to Archard's law. It is basically based on the deformation and adhesion components of friction. Without measuring the surface energies, the deformability of the polymer surface can only be judged by the knowledge of the mechanical properties, but it also requires the knowledge of a large number of mechanical properties.

- How do on-line friction curves handle static and dynamic frictional resistance due to reciprocating motion?

Author Response

Dear Reviewer,

Thank you for the comments and information regarding the necessary corrections of our paper. In reference to your comments on many aspects of the work, changes have been introduced. Below please find the replies to comments:

  1. 4-5 lines long, mostly incomprehensible or incorrect sentences should be rewritten everywhere, or unnecessary connecting texts should be deleted (e.g. lines 64-70)

ANSWER: Thank you for this remark, accordingly we have modified the longest sentences and hope to make it more understandably.

The beneficial  properties of iPP and its composites, particularly low density, relatively low price, simple processing  and a high modification ability are the main inspirations of a wide industrial use.  of an isotactic polypropylene and its copolymers are a high modification ability, By chemical treatment, reaction of composites and by heterogonous nucleation the  desired application properties may be created.

Another important advantage is the possibility of a pro-ecological recycling ability by means of commonly used processing techniques. This effect is mentioned in the literature, for instance the mechanical properties and the reusability of printed circuit boards, [13] formation of materials from recycled PP blends with TPE [14] or recycling of polyolefins blends [15]. The use of recycled  polypropylene for composites with natural fibers [16], or the reuse of industrial products of PP [17] are another benefits of this polymer.

  1. Sentence in line 103-104 ?? – CORRECTED
  1. Table 1.: Unlucky to use the sliding/moving arrows, other markings are required- CORRECTED
  1. Line 169: ... very low content of additives ..... this is not a definition

ANSWER: This description is now changed to: a commercial polypropylene without any nucleating additives

  1. “contain” word is a verb and not a noun (line 189, 194) – CORRECTED
  1. figure 2: bad figure, contrary to the explanation of line 250. The pin is actually a steel sphere. – CORRECTED
  1. figure 3. Not relevant to the content of the article, can be deleted – CORRECTED
  1. Table 3 photos: In the figures, the parts explained in the text later should be marked with some emphasis, identifiable by the textual explanation. – We have introduced some changes in the comments to these figures and hope to make this part more clear and easier to understand.
  1. Figures 5 and 6. A more detailed material structure explanation is required for the observed hardness change.

ANSWER: The structure explanation of the observed phenomena is added.

  1. Friction curves: analysis of running-in and steady state friction is required separately. The phenomenon of running-in is well known in polymer tribology: the polymer transfer film characterizes the formation on the metal surface and the attainment of its equilibrium state. It is strongly related to the surface energy and adhesion of the polymer.

ANSWER:  Thank you for this remark, accordingly we have added some more information.

A tendency of sliding distance dependent stabilization of the steady state COF values for PP samples with various NA’s composition, produced by mold temperature 20oC and 70oC, may be seen on Figure 6.

As it follows from Figure 6, at the beginning of the measurement the coefficient of friction is relatively low and the grows to finally achieve a steady state. This behaviour is typical for polymers [63,64]. Because of the adhesion of the polymer to the steel coun-ter-sample, a thin layer of polymer (so-called transfer film) can be deposited on the oppo-site sample [65]. All of those factors contribute to the changes of the coefficient of friction in function of sliding distance.

Analysing the changes of Coefficient of friction for all samples (Figure 6), it can be seen that for the samples produced at lower mold temperature the stabilization occurs af-ter a short time. In the case of higher mold temperature, the values of COF are noticeable higher and the stabilization time is generally longer. The differences in stabilization time may result from changes of the outer layer properties, like more intense friction induced deformation of the surface, as it will be shown later by discussion of confocal observation of the patch tracks.

  1. Friction analysis is not possible at least, the simplest model e.g. Without regard to Archard's law. It is basically based on the deformation and adhesion components of friction. Without measuring the surface energies, the deformability of the polymer surface can only be judged by the knowledge of the mechanical properties, but it also requires the knowledge of a large number of mechanical properties.

ANSWER: Thank you very much for this remark, but our goal was to assess how the surface deforms depending on the properties of the materials, and not to determine its mechanical properties.

  1. How do on-line friction curves handle static and dynamic frictional resistance due to reciprocating motion?

ANSWER: This method makes it possible to obtain the averaged values in the counter-sample association despite the reciprocating motion. The data is filtered and the values that are taken into account are characterized by constant speed and applied load.

Thank you once more for the very valuable comments and remarks, we hope that these corrections are sufficient and will positively influence the evaluation of the manuscript.

I hope that these changes are sufficient and will positively influence the evaluation of the manuscript.

I am enclosing the Manuscript.

Yours faithfully,
Natalia Wierzbicka

Reviewer 3 Report

The authors prepared a series of samples with different contents of nucleating agents, characterized their morphology and structure, and discussed the relations between structure and properties (tribology and Shore hardness). The research is instereting and meaningful, but there are some crucial issues that need to be addressed before publication.

  1. For injection molded samples, the structure is not isotropic. Especially there are great differences in morphology and structure between the skin and core layers (see related references). For iPP with beta nucleating agent, the content of beta modification is different in different area of its sample, which depends on the shear rate. In this work, the hardness and tribology properties are mainly related to the morphology and structure of the skin. However, the DSC and WXAD results are the average value of the whole sample along the thickness direction.  Therefore, the conclusions are not well supported by the data. The authors should characterize and analysize the structure of sample layer by layer.
  2. The author had better more in-depth discuss the structural formation, especially the competition of nucleating and growth between alpha and beta modifications if existing. 

Author Response

Dear Reviewer,

Thank you for your valuable comments and remarks concerning the necessary corrections of our article. In reference to your comments on many aspects of the work, changes have been introduced. Below please find the replies to comments:

  1. For injection molded samples, the structure is not isotropic. Especially there are great differences in morphology and structure between the skin and core layers (see related references). For iPP with beta nucleating agent, the content of beta modification is different in different area of its sample, which depends on the shear rate. In this work, the hardness and tribology properties are mainly related to the morphology and structure of the skin. However, the DSC and WXAD results are the average value of the whole sample along the thickness direction. Therefore, the conclusions are not well supported by the data. The authors should characterize and analysize the structure of sample layer by layer.

ANSWER: We fully agree with this remark. As it was shown in our previous publications, the structure gradient on the thickness of differently modified iPP is between others on the shear rate distribution dependent. In our recent paper and in paper published several years ago  we have shown how by using specific markers the local flow velocity distribution on the cross section of injection moulded products may be observed. The aim of this paper was to defined the fracture induced effects, which happens on the surface of the sliding material. The WAXS experiments were realized in reflection mode,  thus the  structure information’s, particularly the creation of both polymorphic structures,  correspond to the outer layer, where the interesting friction induced behaviours were observed in discussed.

  1. The author had better more in-depth discuss the structural formation, especially the competition of nucleating and growth between alpha and beta modifications if existing.

ANSWER: Thank you for this remark, demonstrating an important scientific problem by heterogeneous nucleated crystallization of semi-crystalline polymers.  As mentioned above the aim of our paper was to investigate the friction induced deformation of the sliding surface with well-defined local structure, thus the classical crystallographic problems related to the crystallization kinetics,  mentioned in your question, wasn’t the topic of our scientific interests.

We hope that you may accept the answers on the remarks included in the review, and that the explanation we have presented are sufficient and will positively influence the evaluation of the manuscript.

Yours faithfully,

Natalia Wierzbicka

Reviewer 4 Report

The authors reported friction and hardness of injection-molded isotactic polypropylene (iPP). iPP used were modified with compositions of two nucleating agents (NA’s) which are the 1,3:2,4-bi s(3,4-dimetylobenzylideno) sorbitol (DMDBS) creating the alpha-form crystal and a mixture of pimelic acid with calcium stearate (PACS) forming the beta-form crystal. Effect of the NA’s content and molding temperature on friction and hardness of the injection-molded iPP samples was interesting. Comments for this manuscript are listed as follows:

p.4, Figure 1

The experimental plan was written in the text of a paragraph “2.1 Plan of the experiment”. As that plan is simple and common one, it is better to remove Figure 1 from this manuscript. There is no need to have it.

p.5, line 173h

“The 1,3:24-Bis (3,4-dimetylobenzylideno) sorbitol (DMDBS) Millad 3988” should be corrected to “The 1,3:2,4-Bis (3,4-dimetylobenzylideno) sorbitol (DMDBS) Millad 3988s”.

p.5, line 173h

“pimelic acid and calcium stearate (both delivered by Sigma-Aldrich) in a power form” should be corrected to “pimelic acid and calcium stearate (PACS) (both supplied by Sigma-Aldrich) in a power form”. The abbreviation of sample name should be added to the sentence.

p.5, line 189th,

“the most suitable contain is 0.2wt%” should be corrected to “the most suitable content is 0.2 wt%.”

“contain” is a verb. There are the same mistakes in the other sentences, ex. p.6 lines 227-228th.

p.7, Figure 2

There is no need to have Figure 2 because the wear assessment test using a disk is popular. Explanation in the main text is enough to understand the condition of wear assessment test.

p.8, Figure 3

There is no need to have Figure 3 (photo) because explanation in the main text is enough to understand the condition of sample mounting.

p.9, Table 3

The scale bar of 400 nm in each image should be changed to that of 100 nm.

p.9, line 298th and the caption for Table 3

“The microscopic observation in polarized light microscopy MPO)” should be corrected to “The microscopic observation by polarized optical microscopy (POM)”.

p.9, POM images in Table 3

Special resolution in observed POM images is relatively low. It is better to have magnified images of POM to indicate birefringence images of iPP spherulites. Sample thickness for POM observation should be controlled within a few mm.

p.10, line 330th

“The double melting pic” should be corrected to “The double melting peaks”.

p.10, line 340th

“characteristic for pseudo hexagonal phase contains acc. to Turner-Jones” should be corrected, for example, to “characteristic for pseudo hexagonal phase content calculated with Turner-Jones equation”.

p.11, Figure 4

“b-phase content, k” as a label for the y axis should be corrected to “b-phase content, k”.

p.13, line 381th

What is COCF ?

P.13, Figure 7 and Figure 8, p.14, Figure 8

“Coefficient od friction [-]” or “Coefficient of friction [-]” as labels for the y axis of Figures 7, 8 and 9 should be corrected to “Coefficient of friction”.

If the measured data points are discontinuous, it is better to plot those data points with markers like open and filled circles.

In addition, data profiles were densely plotted in Figure 7. It is difficult to see time dependence of the friction coefficient of each sample. The each profile should be clearly shown in figure.

p.14, line 403th

What is COF? The abbreviation of “Coefficient of friction” should be indicated in the main text.

p.16 and 17, Table 5

Scanning and confocal microscopic images for the samples are shown in Table 5. There is no need to have three-dimensional confocal microscopic images in that table because two-dimensional ones are also shown in the same table. Also there is no need to show all the images in the manuscript. Some of them could be used for Table 5. It is better to submit the other images to the Journal as supporting information.

The authors investigated the effect of crystal forms (alpha and beta forms) and their composition on friction and hardness of the neat and modified iPP samples. However, friction and hardness of polymer samples also depend on crystallinity and crystal orientation. Those structure parameters can be obtained by analyses with wide-angle X-ray diffraction profiles. Changes on friction and hardness of iPP by adding the PMBDS and PACS should be explained with not only crystal forms and their composition but also crystallinity and crystal orientation.

Author Response

Dear Reviewer,

Thank you for your valuable comments and remarks concerning the necessary corrections of our article. In reference to your comments on many aspects of the work, changes have been introduced. Below I am sending replies to comments:

  1. 4, Figure 1

The experimental plan was written in the text of a paragraph “2.1 Plan of the experiment”. As that plan is simple and common one, it is better to remove Figure 1 from this manuscript. There is no need to have it.

Thank you for this remark, the experimental plan was removed in the corrected version of the paper.

  1. 5, line 173h

“The 1,3:24-Bis (3,4-dimetylobenzylideno) sorbitol (DMDBS) Millad 3988” should be corrected to “The 1,3:2,4-Bis (3,4-dimetylobenzylideno) sorbitol (DMDBS) Millad 3988s”. -CORRECTED

  1. 5, line 173h

“pimelic acid and calcium stearate (both delivered by Sigma-Aldrich) in a power form” should be corrected to “pimelic acid and calcium stearate (PACS) (both supplied by Sigma-Aldrich) in a power form”. The abbreviation of sample name should be added to the sentence. -CORRECTED

  1. 5, line 189th,

“the most suitable contain is 0.2wt%” should be corrected to “the most suitable content is 0.2 wt%.”- CORRECTED

  1. “contain” is a verb. There are the same mistakes in the other sentences, ex. p.6 lines 227-228th. – CORRECTED
  2. p.7, Figure 2

There is no need to have Figure 2 because the wear assessment test using a disk is popular. Explanation in the main text is enough to understand the condition of wear assessment test. - CORRECTED

  1. 8, Figure 3

There is no need to have Figure 3 (photo) because explanation in the main text is enough to understand the condition of sample mounting. - CORRECTED

  1. 9, Table 3

The scale bar of 400 nm in each image should be changed to that of 100 nm. - CORRECTED

  1. 9, line 298th and the caption for Table 3

“The microscopic observation in polarized light microscopy MPO)” should be corrected to “The microscopic observation by polarized optical microscopy (POM)”. - CORRECTED

  1. 9, POM images in Table 3

Special resolution in observed POM images is relatively low. It is better to have magnified images of POM to indicate birefringence images of iPP spherulites. Sample thickness for POM observation should be controlled within a few mm.

ANSWER: The scale bar of 400 nm in each image was changed to that of 100 nm, the quality of images were improved. Sample thickness for POM observation was controlled within a few 10^-6 m.

  1. 10, line 330th

“The double melting pic” should be corrected to “The double melting peaks”. – CORRECTED

  1. 10, line 340th

“characteristic for pseudo hexagonal phase contains acc. to Turner-Jones” should be corrected, for example, to “characteristic for pseudo hexagonal phase content calculated with Turner-Jones equation”. – CORRECTED

  1. 11, Figure 4

“b-phase content, k” as a label for the y axis should be corrected to “b-phase content, k”. – CORRECTED

  1. 13, line 381th

What is COCF ? – It was a mistake and is now CORRECTED

  1. 13, Figure 7 and Figure 8, p.14, Figure 8

“Coefficient od friction [-]” or “Coefficient of friction [-]” as labels for the y axis of Figures 7, 8 and 9 should be corrected to “Coefficient of friction”. – CORRECTED

  1. If the measured data points are discontinuous, it is better to plot those data points with

markers like open and filled circles.

ANSWER: measured data points are continuous, so data can’t be plot with markers like open and filled circles.

  1. In addition, data profiles were densely plotted in Figure 7. It is difficult to see time dependence of the friction coefficient of each sample. The each profile should be clearly shown in figure.

ANSWER: The way of data’s presentation was improved and plotted  separately. 

  1. 14, line 403th

What is COF? The abbreviation of “Coefficient of friction” should be indicated in the main text. – CORRECTED

  1. 16 and 17, Table 5

Scanning and confocal microscopic images for the samples are shown in Table 5. There is no need to have three-dimensional confocal microscopic images in that table because two-dimensional ones are also shown in the same table. Also there is no need to show all the images in the manuscript. Some of them could be used for Table 5. It is better to submit the other images to the Journal as supporting information.

ANSWER: Table 5 – three-dimensional confocal microscopic images in that table show the depth of deformation, while  two-dimensional ones show only the shape of deformation area. In our opinion, removing any of the studied materials in the table would make it difficult for the reader to obtain the data’s necessary to understand the properties of the material and its deformation process.

  1. The authors investigated the effect of crystal forms (alpha and beta forms) and their composition on friction and hardness of the neat and modified iPP samples. However, friction and hardness of polymer samples also depend on crystallinity and crystal orientation. Those structure parameters can be obtained by analyses with wide-angle X-ray diffraction profiles. Changes on friction and hardness of iPP by adding the PMBDS and PACS should be explained with not only crystal forms and their composition but also crystallinity and crystal orientation.

ANSWER: Thank you for this important remark. Really the crystallinity and the macromolecular orientation are significant values which may be additionally detected  and evaluated from WAXS diffractogram, and defined by the orientation function, as shown in our previous papers. In the case of this research our interest was to study the relationship between local polymorphic structure of the sliding layer and friction induced deformation of this layer.  The macromolecular orientation may be created as a result of flow velocity distribution on the cross section of the molten formed part, or/and as a result of post-processing, what wasn’t the aim of our research. It’s known that particularly high and well defined values of orientation function are usually achieved by post processing. On the contrary, the relatively low value of this function is generally noted by injection moulding processing, so we think that it may influence the frictional behaviour only in a very limited degree. Another problem is that the classical Hermans orientation function evaluation procedure demand WAXS measurements realized in transmission mode, what wasn’t the case in our investigations, and it is practically impossible to realize such measurements for a particular layer integrated at the sample. 

We hope that the changes we have presented are sufficient and will positively influence the evaluation of the manuscript.

Yours faithfully,

Natalia Wierzbicka

Round 2

Reviewer 2 Report

I accept the answers and corrections, however English mistakes and some typo remained.

Reviewer 3 Report

All the conerns are addressed, so it can be published now.